# *Pectobacterium versatile* Bacteriophage Possum: A Complex Polysaccharide-Deacetylating Tail Fiber as a Tool for Host Recognition in Pectobacterial *Schitoviridae*

**DOI:** 10.3390/ijms231911043

**Published:** 2022-09-20

**Authors:** Anna A. Lukianova, Peter V. Evseev, Mikhail M. Shneider, Elena A. Dvoryakova, Anna D. Tokmakova, Anna M. Shpirt, Marsel R. Kabilov, Ekaterina A. Obraztsova, Alexander S. Shashkov, Alexander N. Ignatov, Yuriy A. Knirel, Fevzi S.-U. Dzhalilov, Konstantin A. Miroshnikov

**Affiliations:** 1Shemyakin-Ovchinnikov Institute of Bioorganic Chemistry, Russian Academy of Sciences, 117997 Moscow, Russia; 2Department of Plant Protection, Russian State Agrarian University—Moscow Timiryazev Agricultural Academy, 127134 Moscow, Russia; 3Biological Faculty, Department of Microbiology, Lomonosov Moscow State University, 119991 Moscow, Russia; 4School of Biological and Medical Physics, Moscow Institute of Physics and Technology (MIPT), 141701 Dolgoprudny, Russia; 5Zelinsky Institute of Organic Chemistry, Russian Academy of Sciences, 119991 Moscow, Russia; 6Institute of Chemical Biology and Fundamental Medicine, Siberian Branch of Russian Academy of Sciences, 630090 Novosibirsk, Russia; 7Agrarian and Technological Institute, RUDN University, 117198 Moscow, Russia

**Keywords:** soft rot, *Pectobacterium versatile*, bacteriophage, *Schitoviridae*, phylogeny, polysaccharide, tail fiber, polysaccharide deacetylase

## Abstract

Novel, closely related phages Possum and Horatius infect *Pectobacterium versatile*, a phytopathogen causing soft rot in potatoes and other essential plants. Their properties and genomic composition define them as N4-like bacteriophages of the genus *Cbunavirus*, a part of a recently formed family *Schitoviridae*. It is proposed that the adsorption apparatus of these phages consists of tail fibers connected to the virion through an adapter protein. Tail fibers possess an enzymatic domain. Phage Possum uses it to deacetylate O-polysaccharide on the surface of the host strain to provide viral attachment. Such an infection mechanism is supposed to be common for all *Cbunavirus* phages and this feature should be considered when designing cocktails for phage control of soft rot.

## 1. Introduction

Bacterial diseases of potatoes (*Solanum tuberosum* L.) inflict substantial economic losses on this essential crop. There is a noticeable impact of blackleg (stem rot) of the vegetating plants and soft rot of seed and ware potatoes. Soft rot *Pectobacteriaceae* (SRP), comprising bacterial genera *Pectobacterium* and *Dickeya*, are considered to be the major causative agents of these diseases [1,2]. Although the symptoms of the disease are similar, the genera above are very diverse and include 19 species of *Pectobacterium* and 11 species of *Dickeya* [3,4,5]. Species of SRP are defined based on microbiological traits, virulence to particular host plants, adaptation to certain environmental conditions and, mostly, on genomic features [6]. One of the most abundant SRP in the pathogenesis of potato soft rot is *Pectobacterium versatile* (Pve). This species was previously considered as a strain group within *P. carotovorum* subsp. *carotovorum* [7] and later was elevated as a separate species [8]. After the establishment of the new species, strains of Pve were identified in historical collections [8] and were found as soft rot pathogens throughout the world [9,10,11,12,13,14,15]. Pve is usually not as aggressive toward potatoes as *P. atrosepticum* or *Dickeya solani* [16]. However, it is ubiquitous in fields and the environment and has a broad range of affected host plants. Therefore, this pathogen is a common cause for soft rot in stored tubers and should be considered among the primary target pathogens of potatoes to be monitored and controlled.

Current agricultural practices are experiencing a lack of effective chemical reagents to combat SRP [17]. Biological control of these pathogens to protect potatoes is considered promising, particularly the use of bacteriophages (phages), natural parasites of bacterial populations [18,19]. The requirements specifying the suitability of phages for biocontrol/therapy purposes are defined by the scientific community [20,21,22]. A number of successful applications of phage cocktails to control SRP *in planta* and in the field have been reported [23,24,25].

Very few SRP-specific phages possess a host range broad enough to infect all strains within a bacterial species. As a result, there is coverage of some particular strain groups dependent on the number and diversity of the test isolates, while some strains of the same species remain resistant to the phage.

Among Pve, there is an example, the strain F131, which is not infected by phages such as PP16 [26] and PP99 [27] specific to the majority of Pve strains. Therefore, strain F131 was used as an isolation host to identify infective bacteriophages. These phages, named Possum and Horatius, have been found to belong to N4-like Podoviruses. Phages of this group were previously reported to infect *P. atrosepticum* [23] and *P. parmentieri* [28]. As a result of dramatic changes in the taxonomy of bacteriophages (and viruses in general) [29,30], N4-like phages were established as a separate family, *Schitoviridae* [31].

Combining bacteriophages that are different in their taxonomic attribution and utilizing different receptors on the surface of the bacterial cell are important aspects of phage cocktail formulation. They reduce the risk of the appearance of phage-resistant mutants in the population of pathogenic bacteria [32]. Therefore, the study of the interaction mechanism between novel phages and their host cells was another purpose of the work presented here.

## 2. Results

### 2.1. General Biological Features of Phages Horatius and Possum

Bacteriophages Horatius and Possum were identified as being specifically infective to the strain F131, which is resistant to most other phages specific to Pve. Both phages produce clear plaques 2–3 mm in diameter (with top agar 0.75% *w*/*v*) on their host strain F131. The host ranges of Horatius and Possum are identical, as examined using 52 strains representing various species of *Pectobacterium* and *Dickeya* and other bacteria related to soft rot (Appendix A). Among the strains used in the study, phages Possum and Horatius were able to infect their isolation host Pve F131 only. Both phages did not infect remaining strains of *P. versatile*, as well as any strains of other species (*P. atrosepticum*, *P. polaris*, *P. parmentieri*, *P. carotovorum*, *P. aquaticum*, *D. solani* and *D. dianticola*).

Phages Horatius and Possum have been tested for stability over a wide range of temperatures, pH, ionic strength, chloroform and UV resistance. Both phages demonstrate similar biological properties. Phages are stable in a wide temperature range and survive freezing at −20 °C with no significant loss of titer and heating to 42 °C during a 24 h period with a loss of titer of 30%. Both phages are insensitive to high salinity. The titer after 24 h of incubation at room temperature in 5 M, 1 M, 0.1 M and 0.01 M sodium chloride solution did not differ statistically from the control. Phages were stable in the near-neutral pH range (pH 5 and 7), however, both in acidic (pH = 2) and alkaline (pH = 12) media, phages did not survive. In addition, both Possum and Horatius were found to be sensitive to chloroform. One hour of incubation with chloroform at room temperature caused a 100-fold titer drop. Neither of the phages survive 10 min of UV exposure (25 W) (Appendix A).

The adsorption of both bacteriophages to the isolation host Pve F131 at MOI = 0.001, 28 °C and an aeration mode of 230 rpm occurred relatively slowly. However, after 20 min of incubation, the binding of 100% of phage particles was achieved (Appendix A). The latent period at MOI = 0.01 under the same culture conditions for both phages was 30 min (not including the 20 min adsorption time). The lysis curves of their phages were almost identical. In both cases, lysis led to a phage titer of 1.6 ± 0.2 × 10^8^, which corresponded to the yield of 125 ± 20 progeny particles per cell (Appendix A).

The morphology of bacteriophages Horatius and Possum was studied using transmission electron microscopy. Both phages feature a C1 morphotype with icosahedral capsids 69 ± 3 nm in diameter and short, non-contractile tails 25 ± 2 nm in length (Figure 1). A set of appendages, or whiskers, surrounding the tail, suggests a developed adsorption apparatus of the phages. The virions are slightly bigger than *Autographivirinae* Podoviruses, and the appearance of the particle similar to the previously studied N4-like phages infecting SRP [23,28] suggests the attribution of Horatius and Possum as members of the recently proposed family *Schitoviridae* [31].

### 2.2. Genomic Properties of Phages Horatius and Possum

*Pectobacterium* phages Horatius (GenBank accession #MN812891) and Possum (GenBank #MN812867) have linear dsDNA genomes of 73,737 bp and 73,752 bp, respectively. The genome size and structure are similar to those of *Pectobacterium* phages CB1, CB3 and CB4 [23], phage Nepra [24] and phages ϕA38 and ϕA41 [28], belonging to the N4-like phage group. The genomes of phages Horatius and Possum contain 104 predicted genes and encode 102 proteins and 2 tRNAs (Asn and Gln). Putative functions were assigned for 52 proteins, and 50 gene products were annotated as hypothetical proteins (Appendix A). The genomes are flanked with 636 bp-long terminal repeats. Genomes of these two phages are almost identical, with the exception of a 15 bp-long insert in Possum’s rIIB protector from prophage-induced early lysis gene 49 and a few single nucleotide replacements not affecting the amino acid sequences of putative encoded proteins. Hence, these phages can be assumed to be representatives of the same viral species and thereafter we discuss the features of phage Possum only. The genetic map of *Pectobacterium* phage Possum is shown in Figure 2.

The genes are clustered in several blocks, oriented in opposite directions. The first 60 genes (early region and most replication and transcription genes) are oriented in a forward direction, the next seven genes (encoding tail fibers and a lysis cassette) are in a reverse direction and then the next 15 genes (basically belonging to the DNA topology manipulation group) are oriented forwards again; most of the remaining genes (encoding structural proteins) are oriented in a reverse direction. The GC-content of the Possum genome is 48.5%, distributed unevenly from 44.4% in the early region to about 51% for structural genes. The average value is smaller than the GC-content (52.1%) of bacterial host Pve F131. An analysis of the GC-content of other *Cbunavirus* genomes (47.7–48.6%) also showed values lower than the GC-content of respective hosts’ genomes. This feature seems to be common for some other N4-like phages infecting *Enterobacterales* (e.g., *Klebsiella* phage vB_KpP_FBKp27—44.2%, *Escherichia* phage N4—41.3%, *Escherichia* phage vB_EcoP_G7C—43.3%, etc.). No virulence factors were found in the phage genome.

The genomes of *Schitoviridae* harbor at least three genes for DNA-dependent RNA polymerases [33,34,35]. The hallmark of this phage family is the gene for a large, encapsulated RNAP (vRNAP) [33]. vRNAP is injected into the host cell together with the phage DNA for the immediate start of early gene transcription. Interestingly, recent studies found the encapsulated ejected virion RNAP in other phages with Podoviral morphology distant from N4 [36]. The other two genes encode the subunits of RNA polymerase II, which is necessary for the transcription of middle and late genes [34]. *Pectobacterium* phage Possum possesses all three genes that encode vRNAP (orf83, 11,082 bp), small (orf23) and large (orf25) subunits of RNA polymerase II.

Besides the RNAP II genes, the early-genes region of the Possum genome encodes a dsDNA-binding protein and a number of hypothetical proteins, including membrane proteins, many of which are unique for N4-like *Pectobacterium* phages or for Possum only (Figure 3, Appendix A). Similar to other *Cbunaviruses* [23], the early-genes region of Possum contains a predicted stem loop structure that is typically associated with the N4 early genes promoter. The content of the Possum early genes region corresponds to the assumption that early genes of a N4-like phage genome often encode proteins that interact with the host and encode the components of the middle region transcription machine [37].

The middle-genes region includes replication, phage defense systems and other genes. The phage defense systems can include a host nuclease inhibitor protein, which can protect the viral DNA against host recBCD-mediated degradation [38], DNA methylase, which can protect the phage DNA from host nuclease digestion [39] and an rIIB protector from prophage-induced early lysis protein. The replication apparatus of the Possum genome contains genes including DNA polymerase I, DNA primase, DNA helicase, etc., common for N4-like phages. The block of middle genes containing, basically, replication, DNA topology manipulation and phage defense genes is interrupted by the interposed block of tail fiber and lysis genes oriented in a reverse direction, and the 5′ end of the latter block neighbors the gene for the HNH homing endonuclease. The latter presupposes a recombination event leading to rearrangements in the phage genome. The remaining structural genes located in the late region of the Possum genome encode typical N4-like phage virion protein and the two-subunit packaging terminase, and are also oriented in the reverse direction.

### 2.3. Taxonomy and Phylogeny

In order to elucidate the taxonomic position of *Pectobacterium* phages Possum and Horatius, whole-genome comparisons and phylogenetic studies were carried out. The calculations of the average nucleotide identity (ANI) conducted with the orthoANIu tool [40] pointed to *Pectobacterium* phages vB_PatP_CB1, CD3, CB4, ϕA38 (and almost identical ϕA41) and Nepra as being the closest relatives of Possum, with similar ANI values of about 94% (Appendix A). The VIRIDIC tool is used by the International Committee on Taxonomy of Viruses (ICTV) as a primary classification technique [41]. The VIRIDIC matrix obtained (Appendix A) indicates that all *Pectobacterium* phages listed above are grouped together in one single cluster, with an intergenomic similarity in excess of 70%. This means that all these phages can be considered to be members of the same genus *Cbunavirus* of the *Schitoviridae* family. The *Schitoviridae* family includes most of the phages described as “N4-like viruses” previously, as well as the related phages revealed with genomic analysis and phylogeny [31].

The phylogenetic studies of conservative proteins were conducted using the sequences of major capsid protein (MCP), portal protein (PP), large subunit of terminase (TerL) and their concatenated protein sequences. All the resulting trees indicate the monophyleticity of the branch comprising the Possum, other *Cbunavirus* phages and the *Klebsiella* phage vB_KpP_FBKp27 (Appendix A).

### 2.4. Bioinformatic Analysis of the Adsorption Apparatus and Production of Recombinant Tail Fiber Protein

The phage adsorption complex is critical for host specificity [42,43]. In tailed phages, the adsorption apparatus comprises receptor-binding proteins (RBPs). Usually, tail fibers and/or tail spikes are considered to be primary RBPs. Some reports indicate baseplate proteins [44] and decoration proteins [45] as possible components able to interact with receptor structures on the bacterial surface. Lipopolysaccharides (LPS) or polysaccharide capsules are frequent receptors for tail fiber and tail spike proteins [46,47]. Some of those proteins can depolymerize surface polysaccharides or deacetylate them, leaving the backbone of the polysaccharide intact [48,49].

Mechanisms of host adsorption are highly variable among *Schitoviridae*. *Escherichia* phage N4 uses the host’s outer membrane protein NfrA as the recognition receptor and the “tail sheath protein” (TShP, encoded by orf65) as the receptor-binding protein [50]. TShP forms an apron-like structure [51] around the non-contractile tail. The same type of protein (gp69) is considered to serve as an RBP for *Achromobacter* phage phiAxp-3 [52], but in this case the receptor molecule is an LPS [53]. *Escherichia* phage G7C uses LPS as a receptor, although the adsorption apparatus comprises branched tail fibers (gp63.1 and gp66), where the first one contains the SGNH hydrolase domain and deacetylates the LPS [54]. LPS and outer polysaccharides were shown to serve as a primary receptor for *P. parmentieri* phage φA38 [55] and *Erwinia amylovora* phage S6 [56,57].

Bioinformatic analysis of the gene products of *Pectobacterium* phage Possum conducted with BLAST and HMM searches indicated the presence of two proteins (encoded by orf66 and orf67, Figure 4) with notable homology to the RBPs of other phages, both in primary sequence and predicted structure (Appendix A). The key role in phage adsorption is played by Possum gp66. The central part of this protein bears a domain similar to phage and bacterial SGNH hydrolases (Phyre2 confidence > 95%), homologous to polysaccharide deacetylase from phage G7C [54]. The C-terminal part (amino acid residues 825–925) is similar to receptor-binding domains, particularly to *Listeria* phage RBP (HHpred probability 92.4%). Therefore, the experimental determination of the Possum gp66 role in phage-cell interaction was assigned as a goal of the present study. The N-terminal part of gp66 contains a β-structural “tail spike attachment domain” (aa 90–180), which is typical for the mutual linking of branched tail fibers/spikes like *Escherichia* phages CBA120 [58], K1E/K1-5 [59] and G7C [54]. This part of gp66 is complicated by a predicted α-helical structure (aa 1–90), with unclear function. Previous studies demonstrated that the N-terminal domains of tail fiber/spike proteins are not essential for receptor-binding functions but hinder the stability of recombinant proteins. The goal of correct folding and oligomerization of the deletional mutant of the target protein can be achieved by its fusion with SlyD, a bacterial peptidyl-prolyl cis/trans isomerase with chaperone properties. Chimeric protein can be purified using a unified protocol, and after proteolytic removal of SlyD yields soluble and functional RBPs [60]. This approach using the cloning vector pTSL (GenBank accession KU314761) [60] was successfully applied to obtain fibrous proteins from pyocins [61], and bacteriophage tail spike/fiber proteins [27,61,62]. The same strategy was used in the present work, and the recombinant C-terminal fragment of Possum gp66 (Dgp66) lacking amino acids 1–196 was a trimeric protein stable in solution and demonstrating the binding to host polysaccharide and processing it (see further).

It is also worth to note that the attribution of phage RBPs as “fibres: or “spikes” is comparatively arbitrary (reviewed in [49]). Based on historical definition and the proposed bipartite composition we regard the components of the adsorption apparatus of phage Possum as “fibres”. However, the presence of the enzymatic domain in gp66 and compact predicted architecture may define them as “spikes” as well.

### 2.5. O-polysaccharide of Pectobacterium versatile F131 (VKM-3418)

Sugar analysis of the O-specific polysaccharide (OPS) of *P. versatile* F131 revealed rhamnose (Rha), galactose and glucosamine in the ratios ~2:1.5:1 (GLC detector response), respectively. Further studies by NMR spectroscopy showed that the OPS also includes galacturonic acid (GalA). GLC of the peracetylated (S)-2-octyl rhamonosides [63] indicated that rhamnose has the L configuration. The D configurations of the other constituent monosaccharides was established by analysis of 13C NMR data of the OPS, taking into account known regularities in glycosylation effects [64].

The 13C NMR spectrum of the OPS (Figure 5A, top) shows, *inter alia*, signals for five anomeric carbons at δ 97.8–104.9, one CH3-C group at δ 17.8 (C-6 of Rha), four hydroxymethyl (HOCH2-C) groups at δ 61.0–62.8 (C-6 of Gal and GlcN), one carboxyl (HO2C-C) group at δ 173.9 (C-6 of GalA) and other sugar carbons at δ 69.5–83.1. There were also signals of two N-acetyl groups at δ 23.6–23.7 (CH3) 175.9 and 176.2 (CO), and O-acetyl group at δ 22.0 (CH3) and 174.7 (CO). Accordingly, the 1H NMR spectrum of the OPS (Figure 5B, top) displayed signals for five anomeric protons at δ 4.53–5.37, one CH3-C group at δ 1.27 (3H, d, J5,6 7.2), two N-acetyl groups at δ 1.95 and 2.0.6 and one O-acetyl group at δ 2.25. These data demonstrated that the OPS has a pentasaccharide repeating unit containing two residues of D-GlcNAc and one residue each of L-Rha, D-Gal and D-GalA, all monosaccharides being in the pyranose form. One of the monosaccharides is O-acetylated.

The 1H and 13C NMR spectra of the OPS were assigned (Appendix A) using two-dimensional 1H,1H (COSY, TOCSY, ROESY) and 1H,13C (HSQC and HMBC) experiments (Appendix A). The β configuration of one of the GlcNAc residues (unit D) was established by a relatively large coupling constant of J1,2 7.2 Hz and the α configuration of the other GlcNAc residue (unit A), as well as Gal and GalA residues (units C and E, respectively), by a smaller coupling constant of J1,2 ~3 Hz. The β configuration of unit D was confirmed by H-1/H-3 and H-1/H-5 correlations observed in the 1H,1H ROESY spectrum. The α configuration of Rha residue (unit B) was inferred from the 13C NMR chemical shifts of C-5 (δ 70.5) compared with published data of the corresponding α- and β-rhamnopyranoses [65] and the absence of H-1/H-3 and H-1/H-5 correlations in the 1H,1H ROESY spectrum. These data also demonstrated that all sugar residues occurred in the pyranose form. Downfield displacements to δ 75.4–83.1 of the signals for the linkage carbons in the 13C NMR spectrum of the OPS, namely C-2 of Gal, C-3 of Rha and β-GlcNAc, C-4 of GalA and α-GlcNAc, as compared with their positions in the corresponding non-substituted monosaccharides [64] revealed the glycosylation pattern in the repeating unit. All monosaccharides were found to be monosubstituted as follows: Gal at position 2, Rha and β-GlcNAc at position 3, GalA and α-GlcNAc at position 4. Therefore, the OPS is linear.

The linkages between the monosaccharides were confirmed and their sequence in the repeating unit was established using the 2D 1H,1H ROESY spectrum (Appendix A), which displayed inter-residue correlations between the following anomeric protons and protons at the linkage carbons: α-GlcNAc A H-1/GalA E H, GalA E H-1/β-GlcNAc D H-3, β-GlcNAc D H-1/Gal C H-2, Gal C H-1/Rha B H-3, Rha B H-1/α-GlcNAc A H-4 at δ 4.91/4.36, 5.37/3.72, 4.53/3.78, 5.24/3.99 and 4.97/3.66. Accordingly, the 1H,1C HMBC spectrum (Appendix A) showed the following correlations between anomeric protons and linkage carbons at δ α-GlcNAc A H-1/GalA E C-4, GalA E H-1/β-GlcNAc D C-3, β-GlcNAc D H-1/Gal C C-2, Gal C H-1/Rha B C-3, Rha B H-1/α-GlcNAc A C-4 at δ 4.91/80.5, 5.37/72.6, 4.53/79.5, 5.24/75.4 and 4.97/78.3. A low-field position at δ 5.20 of the signal for H-2 of Rha, as compared with its position at δ 3.92 in the non-substituted α-rhamnopyranoside [66], was evidently due to a deshielding effect of an O-acetyl group [67] and indicated that the Rha residue is 2-|O-acetylated.

Therefore, the OPS has the structure shown in Figure 6. To the authors’ knowledge, this structure is new among known bacterial polysaccharide structures.

### 2.6. Deacetylation of Bacterial O-polysaccharide by Tail Fiber Protein gp66

After treatment of the OPS with Δgp66, a modified polysaccharide (MPS) was isolated. The NMR spectra of the MPS (Appendix A); Figure 6, bottom; Appendix A) were similar to those of the initial OPS but there were two significant differences: (a) the signals of the O-acetyl group had disappeared and (b) the signal for H-2 in the 1H NMR spectrum and the H-2/C-2 cross-peak in the 1H,13C HSQC spectrum of the OPS, which belonged to 2-O-acetylrhamnose, shifted from δ 5.20 and δ 5.20/71.1 to δ 4.23 and δ 4.23/68.3, respectively, i.e., to the position characteristic of a non-O-acetylated rhamnose residue [66]. These data indicated that the MPS has the structure shown in Figure 6. Hence, an interaction of phage Possum’s tail fiber protein Δgp66 with the host O-polysaccharide caused its O-deacetylation. Deacetylase domains as a part of adsorption apparatus, enhancing phage adhesion to the bacteria, were previously reported including those of *Pectobacterium* phages [27]. However, this feature is not typical for N4-like phages with fibrous receptor-binding proteins.

## 3. Discussion

Recent comprehensive taxonomic studies [31] have convincingly classified most Podoviruses related to the “classic” *Escherichia* phage N4 to a new family, *Schitoviridae*, which is further divided into a number of new subfamilies. Phylogenetic studies and intergenomic comparisons indicate that *Pectobacterium* phages Possum and Horatius belong to the genus of *Cbunavirus**es*. The *Klebsiella* phage vB_KpP_FBKp27 seems to be the closest known relative of the *Cbunaviruses* group.

Bioinformatic studies of the genomes of the novel phages reveal features typical of N4-like phages, including the presence of virion RNA polymerase (vRNAP), which has been considered to be a hallmark of the N4-like phage family for a long time. This polymerase belongs to a large category of polyvalent proteins [68], possibly originated by the fission of smaller proteins. The N-terminal part of vRNAP is responsible for protein injection into the host, the central part provides the RNAP activity and a C-terminal is required for encapsidation [51,69]. A sequence search using the protein sequences of *Schitoviridae* vRNAPs and *Autographiviridae* internal core protein with transglycosylase activities reveals distant similarities between these proteins. The internal core transglycosylase (gp16) of phage T7 is encapsidated as four copies [70] and is injected into the host cell similar to N4-like vRNAP. Recent studies [36] have demonstrated the presence of encapsidated vRNAP injected in the host cell in a relatively distant group of phages with Podoviral morphology. These results and structural features of N4-like vRNAP make it possible to hypothesize as to future findings of polyvalent vRNAP-containing phages belonging to other, currently unknown, groups.

The experimental results obtained are consistent with bioinformatic analysis and suggest the participation of a tail fiber protein (gp66) of *Pectobacterium* phage Possum in the adsorption and deacetylation of bacterial O-polysaccharide. The bioinformatic analysis indicated that the tail fiber protein gp66 contained the SGNH domain, which is common for many phage RBP and can be acquired by horizontal transfer. A similarity search suggested that a second, smaller tail fiber protein, Possum gp67, is a component of the adsorption apparatus. This protein has no pronounced receptor-binding domain; however, it is perfectly conserved among *Cbunavirus* phages. It is suggested that it serves as a connector protein where the N-terminus is attached to the virion, and that the a-helical C-terminus provides the link with the α-helical N-terminus of the larger polysaccharide-deacetylating tail fiber. This composition of the RBP complex may be a hallmark of *Cbunavirus* and related phages within *Schitoviridae*.

The biological properties of phages Possum and Horatius are comparable with those reported previously for *Cbunavirus* phages [23]. However, the host range of these phages is narrow, and this is explained by the unique structure of O-polysaccharide serving as a primary bacterial receptor. Therefore, the applicability of *Cbunaviruses* for phage control is limited. The strict correspondence between tail fiber enzymatic activity and the structure of the receptor probably plays an important role in avoiding restrictive strains and in recognizing permissive hosts [71]. Thus, the phages provide the most efficient reproduction, which is possible only in a limited number of strains (e.g., those lacking some defense systems). On the other hand, modular composition of the adsorption apparatus may facilitate the adaptation of phages to new hosts via the lateral transfer of genes encoding functional tail fibers.

Besides numerous financial and legal problems obstructing the implementation of phage control to modern agriculture, and some understudied issues in application protocols, an important point of successful phage control is the construction of comprehensive collections of phages infective to all possible strain diversity of target pathogens. The example of soft rot *Pectobacteriaceae* demonstrates that this diversity is often underestimated. After the formation of numerous new taxons within *Pectobacterium* and *Dickeya* spp., some of the newly formed species are still not monophyletic and are potential subjects for further dissection. Even the taxons established long ago (e.g., *P. atrosepticum* within SRP) may contain “orphan” strain groups with phage susceptibility different from most other representatives of this species. In the case of *P. versatile*, the strains similar to F131 are the examples of such “orphan” strains, which are, nevertheless, abundant among SRPs circulating in Russia [72]. Therefore, the phages like Possum infecting such strains should be included to the prospective cocktails for phage control of SRP despite a narrow host range.

## 4. Materials and Methods

### 4.1. Bacterial Strains

*Pectobacterium versatile* strain F131 was isolated in 1992, in the Leningrad region, Russia, from symptomatic potatoes and initially was attributed as *P. carotovorum* subsp. *carotovorum*. After comprehensive biological characterization and sequencing (NCBI GenBank accession number NZ_PDVW00000000.1) [7], it was deposited into the Russian Collection of Microorganisms as *P. versatile* VKM–3418 and is available on request. This strain was used for the propagation of phages Horatius and Possum. The general properties of other stains of SRP and other soft rot-associated bacteria used to assess the host range of phages are presented in Appendix A. All strains were cultivated on lysogenic broth (LB) liquid or agar medium at 28 °C.

### 4.2. Isolation and Purification of Phages Horatius and Possum

Phages Horatius and Possum were isolated using an enrichment method. River water was collected near the drainage of processed urban wastewater near Moscow, Russia (Possum) and Potsdam, Germany (Horatius). For phage isolation, an environmental water sample (750 mL) was mixed with 4× LB liquid media. Then 1 mL of overnight culture of Pve strain F131 was added and incubated overnight at 28 °C. The culture was centrifuged at 8000× *g* to remove unlyzed bacteria and the supernatant was filtered (0.45 µm pore-size filter, Millipore, Burlington, MA, USA). The supernatant was spotted (10 µL) onto LB overlays that were seeded with Pve F131. Phages were isolated by picking individual plaques, which were then repeatedly plated and titered to ensure purity.

Phages were preparatively propagated using Pve F131 as a host strain, according to a conventional technique [73]. Cell lysate was treated with chloroform, centrifuged to remove cell debris (8000× *g*, 20 min), sterilized by being filtered through a 0.22 mm pore size membrane filter (Millipore) and treated with DNase I (0.5 mg/mL) and RNase for 60 min. Phage particles were pelleted in the ultracentrifuge (100,000× *g*, 60 min, 4 °C, Type 45 rotor, Beckman-Coulter, Brea, CA, USA) and further purified by CsCl step gradient (0.5–1.7 g/mL density range) ultracentrifugation (22,000× *g*, 120 min, 4 °C, SW28 rotor, Beckman). The resulting suspensions of phages were dialyzed overnight against SM buffer (10 mM Tris-HCl, pH 7.4, 10 mM MgSO_4_), to remove CsCl. Purified phages were stored at 4 °C in SM buffer as a transparent, slightly opalescent liquid.

### 4.3. Electron Microscopy

Purified phage particles were applied to grids and stained with a 1% uranyl acetate aqueous solution [74]. The specimens were visualized using a Libra 120 electron microscope (Zeiss, Oberkohen, Germany) at 100 kV accelerating voltage. The dimensions of viral particles were averaged from the readings of at least 20 phages.

### 4.4. Host Range and General Characterization of Phages Horatius and Possum

Host ranges of the phages were assayed by direct spotting phage suspensions (~10^6^ pfu/mL) onto a bacterial lawn. In cases where the plaques were observed, the infectivity of the phages was ensured by standard plaque formation assay using phage serial dilutions. Bacterial strains listed in Appendix A were cultivated on LB agar at 28 °C. In phage adsorption experiments, the host strain was grown to an OD_600_ ~0.30 and infected with individual phages at a multiplicity of infection of 0.001. At 1, 2, 3, 4, 5, 10, 15, and 20 min after infection, 100 mL aliquots of the phage–host mixture were taken and transferred into an 800 mL LB medium supplied with 50 mL of chloroform. After bacterial lysis, the mixtures were centrifuged, and the supernatant was assayed to determine the amount of non-adsorbed or reversibly bound phages. One-step-growth assays were performed according to [73]. To assay the lytic activity of phages, an exponentially growing culture of host bacteria (10^8^ cfu/mL) was mixed with phage suspension (MOI of 0.01). The mixture was then incubated with agitation at 28 °C. Every 5 min, aliquots were taken, chilled, and centrifuged, and the appropriate dilutions of the supernatant containing unbound phages were spread on LB agar plates and incubated overnight at 28 °C. The next day, colonies were counted. All experiments were performed independently 3–4 times, and the results were averaged. Phage stability was studied as described in [75] by incubating a 10^3^ pfu/mL phage suspension in SM buffer at different temperatures (−20, +4, +15, +20, +37, +42, +80 °C) or in a range of buffer solutions (20 mM Tris-HCl/20 mM Na citrate/20 mM Na phosphate), adjusted with NaOH to the pH range 2–12. The effect of UV radiation was tested in sterile SM buffer at room temperature. Phage suspensions were adjusted to 10^3^ pfu/mL and were exposed under a UV lamp (25 W) for 5 and 10 min. Chloroform sensitivity was examined by incubation in SM buffer with the addition of chloroform for 1 h. The effect of salinity was studied in a NaCl solution with a concentration of 0.01, 0.1, 1, and 5 M.

### 4.5. Genome Sequencing and Annotation

Phage DNA was purified by phenol extraction and fragmented with ~600 bp fragments in a microTUBE Adaptive Focused Acoustics (AFA) fiber snap-cap tube using a Covaris S2 instrument (Covaris LLC, Woburn, MA, USA). The DNA library was constructed using the dual-index NEBNext multiplex oligos (New England Biolabs, Ipswich, MA, USA) and the NEBNext Ultra II DNA library prep kit for Illumina (New England Biolabs). The library was size-selected on a Blue Pippin 1.5% agarose DNA gel (Sage Science, Beverly, MA, USA) with size-selection settings of 550–1000 bp. This DNA library was sequenced with reagent kit version 3 (600-cycle) on a MiSeq platform (Illumina, San Diego, CA, USA) at the SB RAS Genomics Core Facility (ICBFM SB RAS, Novosibirsk, Russia). Entire genomes of phages Horatius and Possum were assembled de novo using SPAdes software version 3.11.1, with default parameters [76]. Phage genomes were annotated by predicting and validating open reading frames (ORFs) using Prodigal 2.6.1 [77], GeneMarkS 4.3 [78], and Glimmer 3.02 [79]. Identified ORFs were manually curated to ensure fidelity. Functions were assigned to ORFs using a BLAST search on NCBI databases (http://blast.ncbi.nlm.nih.gov, accessed 15 November 2020) and homology criterion of E-value less than 10^−5^, Phyre2 server (http://www.sbg.bio.ic.ac.uk/~phyre2, accessed 20 December 2020) [80] and similarity criterion of Phyre2 confidence of 99% and higher, InterPro (https://www.ebi.ac.uk/interpro, accessed 24 December 2020) [81], and HHpred (https://toolkit.tuebingen.mpg.de/#/tools/hhpred, accessed on 24 December 2020) [82] and similarity criterion of HHpred probability of 99% and higher, using databases PDBmmCIF70, SCOPe70_2.07, ECOD_F70, Pfam-A_v33.1, COG_KOG_v1.0, NCBI_Conserved_Domains_v3.18, SMART_v6.0, TIGRFAMs_v15.0, PRL_v6.9_UniProt_Swiss_Prot_viral70. tRNA coding regions were identified with tRNAscan-SE [83] and ARAGORN [84]. The resulting genomes were visualized using Geneious Prime, version 2021.0.3 (https://www.geneious.com accessed 5 June 2020). The position and length of terminal repeats were identified by searching a region with roughly double the read depth in comparison to the average read depth across the whole genome of the phage and by similarity with terminal repeats of previously reported N4-like phages infecting *Pectobacterium*. Annotated genomes of phages Horatius and Possum have been deposited in the NCBI GenBank under accession numbers MN812691 and MN812687, respectively.

### 4.6. Phylogeny and Taxonomy Studies

Phage genomes were downloaded from the NCBI GenBank (ftp://ftp.ncbi.nlm.nih.gov/genbank, accessed 10 April 2021). Genes of phage major capsid protein, portal protein, and a terminase large subunit were extracted from the annotated genomes. For some unannotated sequences, ORFs were found using Glimmer 3.02. ORFs were validated and corrected by comparison with known homologous genes. Protein alignments were made with MAFFT (L-INS-i algorithm, BLOSUM62 scoring matrix, 1.53 gap open penalty, 0.123 offset value) [85]. The alignments were trimmed manually and with trimAL (http://trimal.cgenomics.org/, accessed 1 May 2021) with -gappyout settings. Phylograms were generated based on the amino acid sequences of proteins and their concatenated alignments. Best protein models were found with MEGAX 10.0.5 [86]. Trees were constructed using MrBayes [87,88]. The robustness of the trees was assessed by the estimation of the average standard deviation in split frequencies.

### 4.7. Genome Comparison, Gene and Protein Analysis

Average nucleotide identity (ANI) was computed using the orthoANIu tool, employing USEARCH (http://www.drive5.com/usearch/, accessed 15 July 2021) over BLAST (https://www.ezbiocloud.net/tools/orthoaniu accessed 20 July 2021) [40] with default settings. The VIRIDIC server (http://rhea.icbm.uni-oldenburg.de/VIRIDIC accessed 20 July 2021) was employed for calculating phage intergenomic similarities (BLASTN parameters ‘-word_size 7 -reward 2 -penalty -3 -gapopen 5 -gapextend -2) [41]. Genome comparison was made with Easyfig [89]. The protein domain search was conducted with InterPro (http://www.ebi.ac.uk/interpro accessed 25 July 2021) [81]. Protein remote homology detection, 3D structure prediction, and template-based homology prediction were carried out using HHpred (https://toolkit.tuebingen.mpg.de/tools/hhpred accessed 25 July 2021) and Phyre2 (http://www.sbg.bio.ic.ac.uk/~phyre2 accessed 25 July 2021) [80]. Custom BLAST databases were mounted with the BLAST tool (https://blast.ncbi.nlm.nih.gov/ accessed 25 July 2021).

### 4.8. Tail Fiber Protein Cloning, Expression, and Purification

The deletional mutant of gene product (gp) 66 of phage Possum (comprising amino acid residues 197–925, Dgp66), which is identical to the corresponding sequence of phage Horatius, was PCR-amplified using primers 5′-ATAGGATCCGGTACTACCAATGGTAATACTG (forward) and 5′-ATAAAGCTTACTGAATAATTCCAGCGTTTC (reverse) with generated BamHI and HindIII cloning sites, respectively. The amplified gene was cloned to vector pTSL using a NEBuilder HiFi DNA Assembly kit (New England Biolabs). Insert-positive clones were selected by PCR, using the same primers, endonuclease hydrolysis, and verified by DNA sequencing. Protein expression was performed in *E. coli* B834(DE3) at 16 °C overnight after induction with 1 mM IPTG. Cells were centrifuged at 4000× *g*, resuspended in a 20 mM Tris-HCl (pH 8.0), 200 mM NaCl buffer, and disrupted by ultrasonic treatment (Virsonic, VirTis, Stone Ridge, NY, USA), and the cell debris was removed by centrifugation at 13,000× *g*. The protein product (Possum Dgp66) was purified on a Ni-NTA Sepharose column (5 mL, GE Healthcare, Chicago, IL, USA) by 0–200 mM imidazole step gradient in 20 mM Tris-HCl (pH 8.0), 200 mM NaCl. The resulting eluate containing the purified protein was dialyzed against 20 mM Tris-HCl (pH 8.0) to remove imidazole, and 6× His-tag was removed by TEV protease (12 h at 20 °C incubation). The target protein was finally purified on a 5 mL SourceQ 15 (GE Healthcare) using a linear gradient of 0–600 mM NaCl in 20 mM Tris-HCl (pH 8.0). Protein concentration was determined spectrophotometrically at 280 nm, using a calculated molar extinction coefficient of 125,250 M^−1^ cm^−1^. The oligomeric state of Possum Dgp66 was assessed by gel-filtration on a calibrated Superdex 200 10 ×300 mm column (GE Healthcare).

### 4.9. Isolation and O-deacetylation of the O-polysaccharides

Lipopolysaccharide (LPS) of Pve F131 was isolated from bacterial cells using the phenol-water method [90]. Contaminating nucleic acids and proteins were precipitated off with aq 50% CCl_3_CO_2_H [91]. An O-polysaccharide (OPS) sample was obtained by degradation of the lipopolysaccharide (112 mg) with aqueous 2% HOAc for 1.5 h at 100 °C. A lipid precipitate was removed by centrifugation (13,000× *g*, 20 min), and the supernatant was purified by gel-permeation chromatography using a 70 × 3.0 cm Sephadex G-50 Superfine (Amersham Biosciences, Uppsala, Sweden) column, using 0.05 M pyridinium acetate buffer pH 4.5 as eluent and monitoring with a differential refractometer (Knauer, Berlin, Germany) to give a high-molecular-mass OPS sample.

The effect of the phage Possum tail fiber protein was studied by the addition of a 300 mg aliquot of Dgp66 to the O-polysaccharide sample (17 mg) and incubation for 2 h at room temperature. The product (9 mg) was isolated by gel-permeation chromatography, as described above.

### 4.10. Sugar Analysis

Hydrolysis of an OPS sample (0.5 mg) was performed with 2 M CF_3_CO_2_H (120 °C, 2 h), and the monosaccharides were analyzed by gas–liquid chromatography (GLC) as alditol acetates [92] on a Maestro (Agilent 7820, Santa Clara, CA, USA) chromatograph equipped with an HP-5 column (0.32 mm × 30 m) (Interlab, Moscow, Russia), using a temperature program of 160 °C (1 min) to 290 °C at 7 °C/min. The absolute configuration of rhamnose was established by GLC of the acetylated (S)-2-octyl glycosides [63] under the same conditions as used in sugar analysis.

### 4.11. NMR Spectroscopy

Samples were deuterium-exchanged by freeze-drying from 99.9% D_2_O. NMR spectra were recorded for solutions in 99.95% D_2_O at 30 °C on a Bruker Avance II 600 MHz spectrometer (Bruker, Bellerica, MA, USA) with a 5 mm broadband inverse probe head for solutions in 99.95% D_2_O at 55 °C for the OPS or 50 °C for the MPS. Sodium 3-(trimethylsilyl)propanoate-2,2,3,3-d_4_ (δ_H_ 0, δ_C_ –1.6) was used as an internal reference for calibration. Two-dimensional NMR spectra were obtained using standard Bruker software, and the Bruker TopSpin 2.1 program was used to acquire and process the NMR data. A spin-lock time of 60 ms and mixing time of 200 ms were used in TOCSY and ROESY experiments, respectively. A two-dimensional ^1^H,^13^C HMBC experiment was recorded with a 60 ms delay for the evolution of long-range couplings to optimize the spectrum for coupling constant *J*_H,C_ 8 Hz.

## Figures and Tables

**Figure 1 ijms-23-11043-f001:**
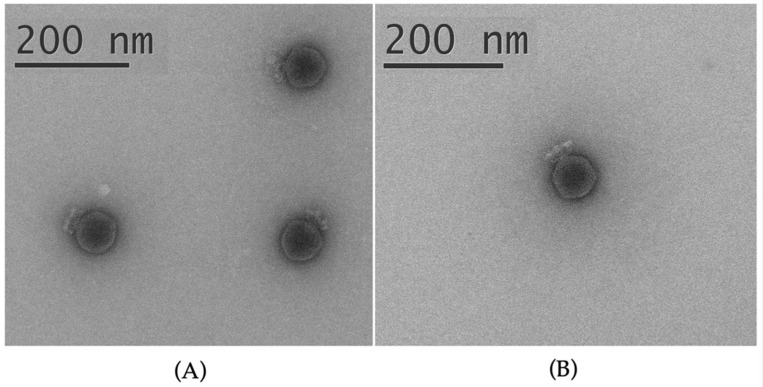
Electron micrograph of the virions of phages Possum (**A**) and Horatius (**B**). Specimens were stained with 1% uranyl acetate.

**Figure 2 ijms-23-11043-f002:**
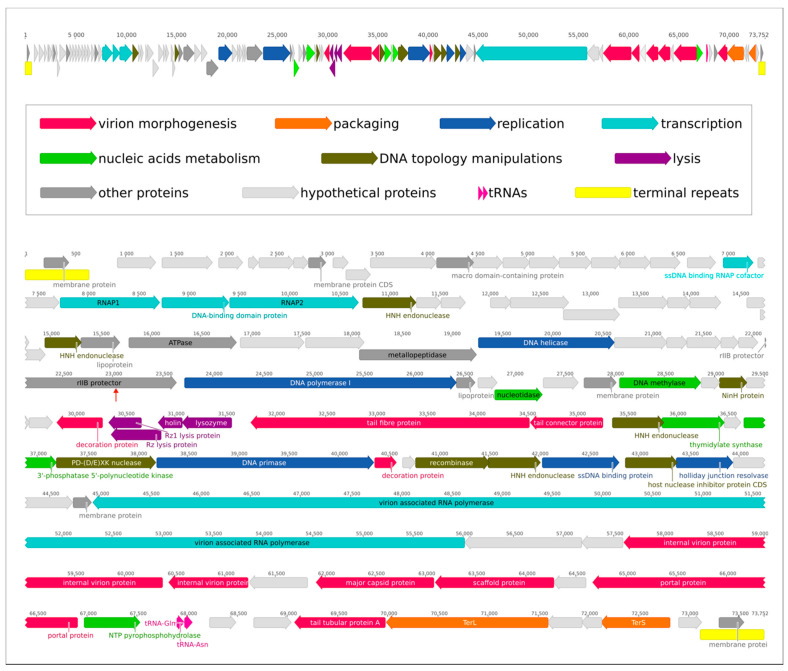
Genetic map of *Pectobacterium* phage Possum. Upper panel: the basic scheme of the genome. Lower panel: the genetic map and putative functions of genes. The 15 bp-long insert in Possum’s rIIB protector from prophage-induced early lysis gene is indicated by a red vertical arrow.

**Figure 3 ijms-23-11043-f003:**
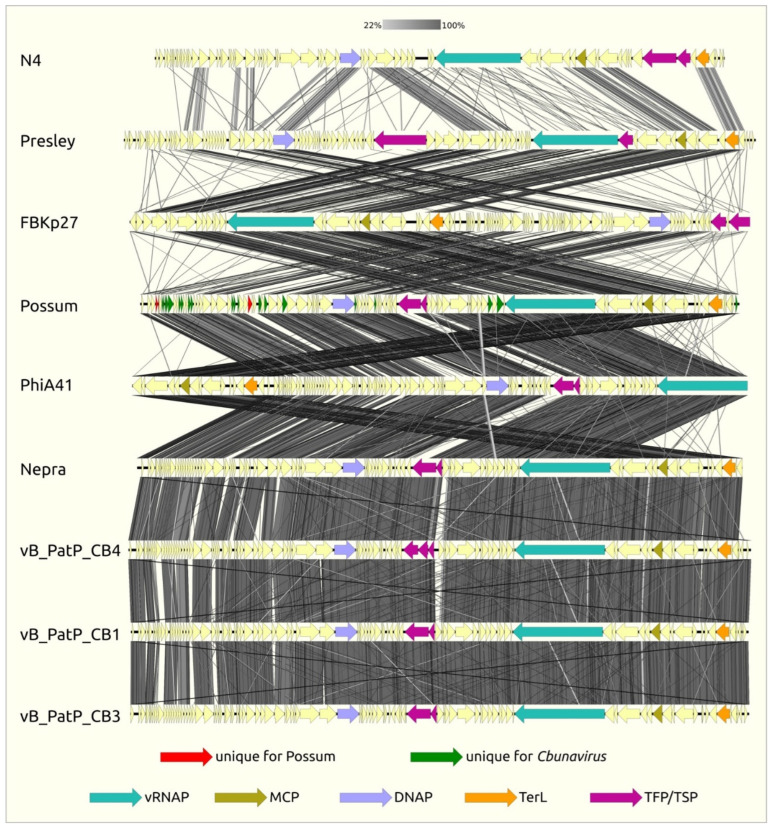
Genome sequence comparison among nine *Schitoviridae* genomes exhibiting co-linearity detected by TBLASTX. The percentage of sequence similarity is indicated by the intensity of the gray color. Vertical blocks between analyzed sequences indicate regions with at least 22% similarity. Genes’ names are as follows: vRNAP, virion RNA polymerase; MCP, major capsid protein; DNAP, DNA polymerase; TerL, large subunit of terminase; TFP/TSP, tail fiber/tail spike protein.

**Figure 4 ijms-23-11043-f004:**
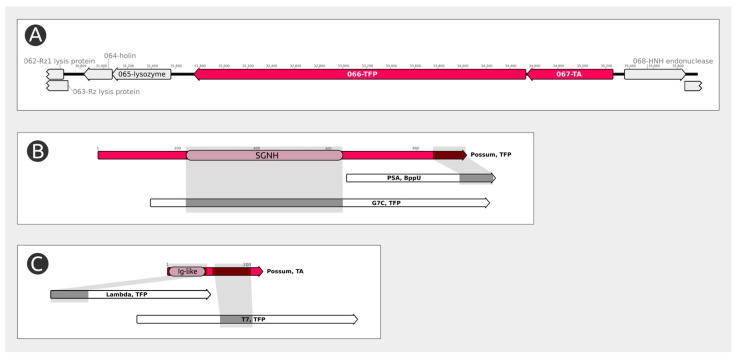
(**A**) The *Pectobacterium* phage Possum genomic map of the region containing the tail fiber genes. (**B**) Schematic representation of structural similarity between the Possum’s tail fiber protein 1 (gp66) and other phages’ proteins obtained with the HHpred HMM-HMM analysis. (**C**) Schematic representation of structural similarity between the Possum’s tail fiber protein 2 (gp67) and other phages’ proteins obtained with the HHpred HMM-HMM analysis. Abbreviations are as follows: TA, tail attachment protein; TFP, tail fiber protein; PSA, *Listeria* phage PSA; BppU, baseplate upper protein; G7C, *Escherichia* phage vB_EcoP_G7C; T7, *Escherichia* phage T7; Lambda, *Escherichia* phage λ; SGNH, region containing SGNH domain.

**Figure 5 ijms-23-11043-f005:**
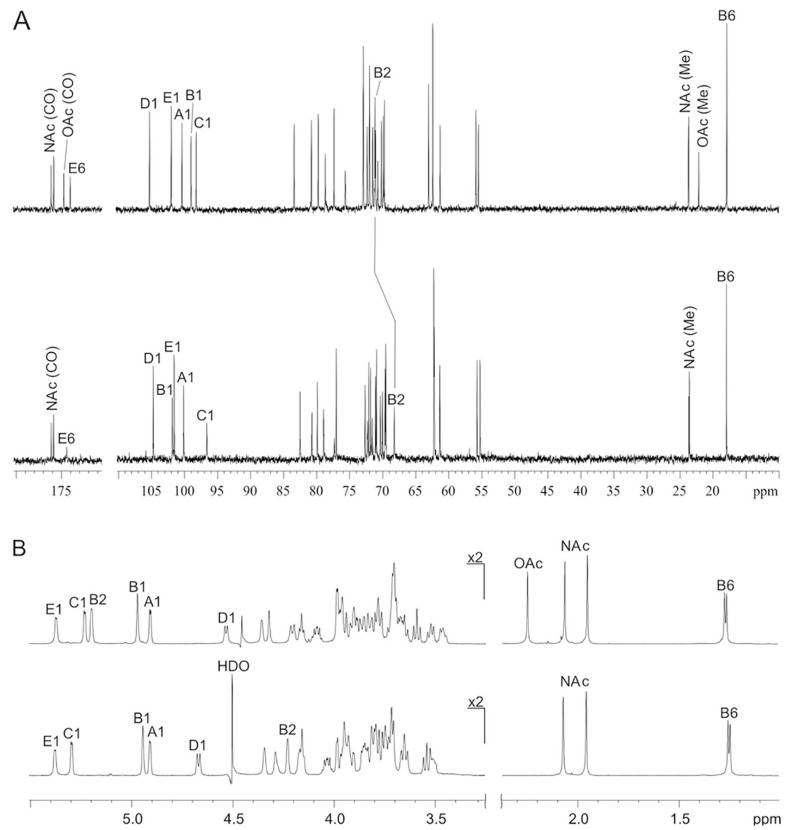
13C NMR (**A**) and 1H NMR (**B**) spectra of the O-specific polysaccharide (**top**) and modified (O-deacetylated) polysaccharide (**bottom**) from *P. versatile* F131.

**Figure 6 ijms-23-11043-f006:**
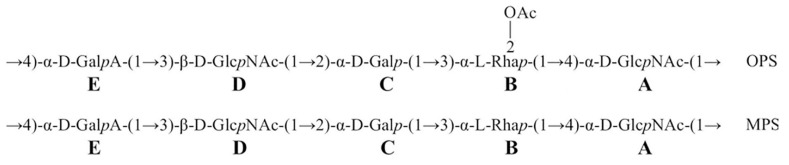
Structures of the O-specific polysaccharide (OPS) and modified (O-deacetylated) polysaccharide (MPS) from *P. versatile* F131.

## Data Availability

Annotated genomic sequences of *Pectobacterium* phages Possum and Horatius have been deposited to NCBI GenBank and are available under accession numbers MN812867 and MN812891, respectively.

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
