# Peer review of "Pectobacterium versatile Bacteriophage Possum: A Complex Polysaccharide-Deacetylating Tail Fiber as a Tool for Host Recognition in Pectobacterial Schitoviridae"

_ijms, 2022, doi:10.3390/ijms231911043_

Round 1
Reviewer 1 Report (New Reviewer)
I consider this manuscript suitable for publication, after the authors address the minor issues outlined below:
- Lines 55: you should put after the term “bacteriophages” (phages) as you mention this short term in line 56. I advise on choose one of the terms (phage or bacteriophage) and use the same throughout the manuscript.
In supplementary figures S1 and S2:
- - In S2 the graphs are not identified as A and B as in the text of the manuscript.
- -The legend of the yy-axe is different in figure S1 and S2. Please be consistent and use the same yy-axe legend for both.
- -The scale of the figures should also be the same, if in S2 you put the PFUs x103 or x107, do the same in figure S1.
- - In S1 the graphs have no lines and in S2 they have lines in the axes, make all graphs the more similar as possible.
- - In S2 the xx-axe legend is “time, min” and in S1 is “t, min”. Be consistent.
Author Response
Thank you for effort in reviewing, the estimation of our work, as well as for valuable comments! We have made corrections in accordance with the comments and suggestions. For Figures S1 and S2, we have unified legends and axis design in both figures and corrected other deficiencies noted.
Reviewer 2 Report (New Reviewer)
This manuscript describes the characterisation of two phage able to infect a bacterial phytopathogen causing disease in potato. DNA analysis shows that the two phage are almost identical.
The authors report infection profiles, phage DNA sequence analysis, phage stability and purification and enzymatic activity of a receptor binding domain fragment.
The manuscript is very clearly written, the methods described in detail and the conclusions are sound. Even though the phage characterised here have a very specific infection profile, it is important to have well characterised phage documented for phage-based infection control strategies.
I had only minor suggestions:
Page 11 section 2.6. Figure numbers are incorrect. Line 339 – should refer to Fig. 6, not figure 7.
Similarly, line 346, should refer to Figure 6, not figure 8.
The number of references (currently 100) could be reduced.
Figure 4 could be re-designed for clarity, and to show some of the data regarding the protein purification. For example, part C of Figure 4 is perhaps not necessary. However, a panel showing for example SDS-PAGE following protein purification and the result of the size exclusion experiment demonstrating the trimeric nature of the delgp66 protein, would be valuable.
Figure S1 – spelling of chloroform.
Having supplementary figure legends alongside the supp figures (as opposed to having them within a single large block of text in the main paper) would be helpful for the reader.
Author Response
Page 11 section 2.6. Figure numbers are incorrect. Line 339 – should refer to Fig. 6, not figure 7.
Similarly, line 346, should refer to Figure 6, not figure 8.
Thank you for your comments. We have made corrections to the text.
The number of references (currently 100) could be reduced.
We find the references relevant to the presented information. We have tried to reduce the number removing the references with duplicating information
Figure 4 could be re-designed for clarity, and to show some of the data regarding the protein purification. For example, part C of Figure 4 is perhaps not necessary. However, a panel showing for example SDS-PAGE following protein purification and the result of the size exclusion experiment demonstrating the trimeric nature of the delgp66 protein, would be valuable.
The protocol of purification of recombinant tail spike proteins using SlyD with subsequent chaperone removal is fairly uniform and was reported many times. The purity of the resulting protein does not play much role in further enzymatic processing, so the gels are not perfect for presentation. Therefore, we would prefer not to present these graphic supplements
Figure S1 – spelling of chloroform.
Corrected
Having supplementary figure legends alongside the supp figures (as opposed to having them within a single large block of text in the main paper) would be helpful for the reader.
Thank you for your comment. We've added descriptions to the supplementary figures
Reviewer 3 Report (New Reviewer)
The manuscript titled "Pectobacterium versatile bacteriophage Possum: A complex polysaccharide-deacetylating tail fiber as a tool for host recognition in Pectobacterial Schitoviridae" refers to newly isolated bacteriophages specifically infecting only their isolation host Pve F131. The Authors present a complete characterization of these two phages starting with General biological features, morphology genomic properties, and taxonomy. They focused on bioinformatics analysis of the adsorption apparatus and assessed the substrate specificity during the infection. They analyzed what changes occur after the phage adhesive protein's treatment of the o-polysaccharide.
The observation was the phages cause deacetylation of O-polysaccharide. The Authors showed this clearly and in detail. Currently, there are some papers reporting on phages with the deacetylation of polysaccharides features. This manuscript excellent extends the knowledge. However, I would like to ask the Authors to add more information about the deacetylation consequences. What exactly is the mechanism, and what it is for? How the mechanism facilitates infecting bacteria?
Author Response
We thank you for the positive assessment of the work we have done and your review. The role of deacetylation in the infectious process is discussed in lines 387-397.
Reviewer 4 Report (New Reviewer)
The authors have isolated two closely related phages against Pectobacterium versatile, which causes potato blackleg, and extensively characterised them. They have also identified the role of their tail fibre protein in the adsorption and deacetylation of bacterial polysaccharides.
I have only a few comments:
Could authors explain their strategy for annotating phage genomes as multiple approaches and databases have been used, but it is unclear how they decided on the results produced at the end?
Line 87, I think the word resistant is inappropriate here as this might be interpreted as these strains were susceptible to these phages before. I suggest the authors rewrite this line, e.g., the isolated phage did not infect strains....
Line 99, please indicate what doses of UV were used.
Figure 2, I suggest authors show the position of promoters and terminators on the genome.
Line 462, do the authors means 3-4 biological replicates?
Author Response
The authors have isolated two closely related phages against Pectobacterium versatile, which causes potato blackleg, and extensively characterised them. They have also identified the role of their tail fibre protein in the adsorption and deacetylation of bacterial polysaccharides.
We sincerely thank you for reading our article, your review and suggestions for improving the text.
I have only a few comments:
Could authors explain their strategy for annotating phage genomes as multiple approaches and databases have been used, but it is unclear how they decided on the results produced at the end?
Thank you for this comment. We've added the explanation in Methods.
Line 87, I think the word resistant is inappropriate here as this might be interpreted as these strains were susceptible to these phages before. I suggest the authors rewrite this line, e.g., the isolated phage did not infect strains....
We have rephrased this sentence in accordance with your recommendation.
Line 99, please indicate what doses of UV were used.
This information is specified in materials and methods (line 470). However, for readability, we have added a clarification to the test results
Figure 2, I suggest authors show the position of promoters and terminators on the genome.
The identification of promoters for N4-like polymerase is not as straightforward as for T7-like phages. Besides the consensus sequence the putative secondary structure of the resulting RNA plays a role, as mentioned in references 33-35. Therefore, we would prefer not to indicate the positions of regulatory sequences since we are not confident about that.
Line 462, do the authors means 3-4 biological replicates?
Yes exactly
This manuscript is a resubmission of an earlier submission. The following is a list of the peer review reports and author responses from that submission.
Round 1
Reviewer 1 Report
Review: Pectobacterium versatile bacteriophage Possum: A complex polysaccharide-deacetylating tail fibre as a tool for host recognition in Pectobacterial Schitoviridae
Authors present bioinformatic and classification for Pectobacterium versatile bacteriophage Possum. They offer sequence analysis and clade relationships for this bacteriophage as well as describe a potential model for phage adsorption via deacetylation of bacterial host cell surface O-polysaccharide.
Introduction:
Elaborate on introduction to include type of hosts, global impact such as economic impact or food supply limitations.
Line 54 – what are current controls on this pathogen?
Line 63 – the limited host range is a shortfall for use broadly as a therapy on as a control mechanism for pathogen
Section 2.2 – authors stress relatedness of Horatius and Possum – state 15bp difference and then only focus on Possum – how relevant is the 15bp to functionality? Could indicate deletion in figure 2.
Is figure 3 needed to support conclusions of this manuscript? If authors deem necessary, they should elaborate via purpose of figure in discussion.
Section 2.3 – Authors address criticism and justify rationale use of VIRIDIC to estimate similarities. This is needed based on their conclusion and their cluster analysis.
Section 2.4, line 293 – need to include figure for the statement that this study demonstrated the deacetylation.
S8 and S9 support authors hypothesis that pg66 and pg67 act as modulators – including could strengthen hypothesis. Could be included in paper.
Organization - line 322 (American English spelling)
Section 2.6 – unclear lead in to experiment performed. Elaborate before conclusion presented. Detailed in M&M but more information needed before results are presented.
Discussion – succinct and clear
Author Response
The Authors would like to thank the Reviewers for careful reading of this manuscript and for constructive comments and suggestions, which help to improve the quality of this paper. We accepted most of the changes they’ve suggested, and fixed the errata they pointed out. However, some of the suggestions remained unchanged, and our points about these is discussed further.
Reviewer 1
Elaborate on introduction to include type of hosts, global impact such as economic impact or food supply limitations.
The economic impact of soft rot is summarized in numerous reviews and even books. We refer Plant Diseases Caused by Dickeya and Pectobacterium Species ISBN 978-3-030-61458-4 where the detailed information is provided. Chapters within this book are referred as 1, 4 and 17
Line 54 – what are current controls on this pathogen?
In lines 56-62 we refer the book mentioned above. Currently, the soft rot management is mainly based on seed certification, hygienic measures and maintaining optimal storage temperatures. All these measures reduce the damage, but they do not provide absolute protection. Therefore the demand in new approaches for pectobacterial treatment is noticeable.
«Current agricultural practices are experiencing a lack of effective chemical reagents to combat SRP [17]. Biological control of these pathogens to protect potatoes is considered promising, particularly the use of bacteriophages, natural parasites of bacterial populations [18,19]. The requirements specifying the suitability of phages for biocontrol/therapy purposes are defined by the scientific community [20–22]. A number of successful applications of phage cocktails to control SRP in planta and in the field have been reported [23–25].»
Line 63 – the limited host range is a shortfall for use broadly as a therapy on as a control mechanism for pathogen
The limited host range of each individual phage can be compensated by using a mixture of phages or a phage cocktail. The total infectious spectrum is thus broad enough to use such a mixture in practice and shows promising results (10.1093/femsle/fnz101, 10.1371/journal.pone.0230842, 10.3390/v13061095). In the presented paper we focus on phages infective to one of such “orphan” P. versatile strains P131 which can be found among circulating strains, but is resistant to other Pectobacterial phages.Thus, Possum is considered as a “complimentary” phage in the cocktail.
Section 2.2 – authors stress relatedness of Horatius and Possum – state 15bp difference and then only focus on Possum – how relevant is the 15bp to functionality? Could indicate deletion in figure 2.
Thank you, we indicated the gene’s number and function in the text of the manuscript. This is a rIIB protector from prophage-induced early lysis gene. We did not conduct special studies to clarify the role of this deletion.
Is figure 3 needed to support conclusions of this manuscript? If authors deem necessary, they should elaborate via purpose of figure in discussion.
Current revisions in phage (and viral in general) taxonomy require thorough study to validate an attribution of the studied phage to a particular taxon. Genome comparison with indicated positioning of hallmark genes is a standard analysis and it was commented in the Results section and mentioned in discussion. This analysis indicates the similarity of Possum and Cbunavirus genomes.
Section 2.3 – Authors address criticism and justify rationale use of VIRIDIC to estimate similarities. This is needed based on their conclusion and their cluster analysis.
We removed this part to avoid unnecessary thorough comparison of different techniques, which is not the primary aim of the study. The VIRIDIC software is now a tool recognized by the International Committee on Taxonomy of Viruses (ICTV) as a primary classification technique.
Section 2.4, line 293 – need to include figure for the statement that this study demonstrated the deacetylation.
Thank you for your valuable note. Indeed, the Figure 8 could be suitable here. However, this would ruin the logic of the narration “genome – protein – function”. Therefore, we reformulated the statement with the bias to the experiment described below in the text.
S8 and S9 support authors hypothesis that pg66 and pg67 act as modulators – including could strengthen hypothesis. Could be included in paper.
We wished to outline the proposed bipartitional composition of the adsorption apparatus of Possum. This feature was not considered as typical for N4-like (Schitoviridae) phages. However, according to the right notice of Reviewer 2 that this observation is somewhat speculative, bioinformatic based and requires further experimental proof, we have shortened the discussion of this matter. We would prefer to leave the figures as supplements which confirm our proposition but not belong to the major findings of the paper.
Organization - line 322 (American English spelling)
The proofreading service we cooperate is resided in the UK, and they use British English (“colour”, “harbour”, “fibre” etc). The rules of the journal permits both ways, requiring the spelling to be uniform ?
Section 2.6 – unclear lead in to experiment performed. Elaborate before conclusion presented. Detailed in M&M but more information needed before results are presented.
NMR spectrum shows that the polysaccharide treated with Possum gp66 is missing the OAc group in all units. Thus it is reasonable to propose that the protein removes this group, being a deacetylase. We have added an explanation clarifying the importance of this finding.

Reviewer 2 Report
The manuscript by Lukianova et al. characterizes newly isolated podophages, Possum and Horatius, that infect a strain of a potato pathogen, Pectobacterium versatile. Phages for this pathogen had not previously been isolated. The manuscript also characterizes the host’s polysaccharide, presumably from the lipopolysaccharide layer of Pectobacterium versatile, with the finding that tail fibers (gp66) of Possum digest this polysaccharide. That is the interesting part of the manuscript, which is generally well written. However, most of the manuscript is dedicated to the relatively uninteresting topic of exactly how to classify the newly isolated phages. This is done with genome sequence-based informatics and can provide material for endless, in my opinion useless, debates. I think that the discussion of classification should be dramatically shortened so that the manuscript focuses on the host polysaccharide, its digestion by gp66 and how much of this is new. The manuscript varies dramatically in significance, interest and overall merit. Concerning the NMR that is used for the polysaccharide characterization, only an expert can understand the details of this important section. I am not the expert needed.
Some relatively minor points:
104: How much UV?
112: The number of cells per ml and the phage titers are very low.
116: The stated length of the tail does not look correct, based on the EM.
117: Not a sentence; also a verb is missing in 119.
209: comparting? Do you mean comparing?
293: State where in this study this information resides?
Not so minor:
310-330: This is prime text for removal as speculation without much point to the speculation.

Author Response
The Authors would like to thank the Reviewers for careful reading of this manuscript and for constructive comments and suggestions, which help to improve the quality of this paper. We accepted most of the changes they’ve suggested, and fixed the errata they pointed out. However, some of the suggestions remained unchanged, and our points about these is discussed further.
The manuscript by Lukianova et al. characterizes newly isolated podophages, Possum and Horatius, that infect a strain of a potato pathogen, Pectobacterium versatile. Phages for this pathogen had not previously been isolated. The manuscript also characterizes the host’s polysaccharide, presumably from the lipopolysaccharide layer of Pectobacterium versatile, with the finding that tail fibers (gp66) of Possum digest this polysaccharide.
That is the interesting part of the manuscript, which is generally well written. However, most of the manuscript is dedicated to the relatively uninteresting topic of exactly how to classify the newly isolated phages.
This is done with genome sequence-based informatics and can provide material for endless, in my opinion useless, debates. I think that the discussion of classification should be dramatically shortened so that the manuscript focuses on the host polysaccharide, its digestion by gp66 and how much of this is new. The manuscript varies dramatically in significance, interest and overall merit. Concerning the NMR that is used for the polysaccharide characterization, only an expert can understand the details of this important section. I am not the expert needed.
We consider that the current changes in phage taxonomy stimulates additional rigor in genome processing. Nowadays attribution of the phage to some particular taxon is almost completely based on genome analysis. Merging the N4-like phages to the separate family is quite a recent event, and the annotations of previously published phages of this group differ in quality, genomic “zero point” selection etc. Thus, we put a substantial effort to genome-based bioinformatics for proper attribution and phylogeny of Possum and Horatius and focusing on hallmark features of Schitoviridae. We have shortened the discussion of available pipelines and the speculative hypothesis about the composition of the adsorption apparatus. But, in general we think the bioinformatic part to be essential and useful for the readers. In order not to overload the text we have provided illustrations concerning the particular bioinformatic computing as supplements.
The NMR part was performed by the experts who specialize on polysaccharide structures. Based on many publications concerning the subject, the provided information is sufficient. Fine details are given as supplementary files.
Some relatively minor points:
104: How much UV?
25W. Thank you for the clarification, added to the materials and methods
112: The number of cells per ml and the phage titers are very low.
We used the standard protocol to measure the biological properties of phages. The phage titer adjusted to MOI and the production yield of ~120 phages per cell look fairly routine.
116: The stated length of the tail does not look correct, based on the EM.
Please see the picture in the attached file where we divided the EM scale bar of 200 nm to 8 resulting in 25 nm, and aligned it with the phage tail.
117: Not a sentence; also a verb is missing in 119.
Changed to “suggests”, thank you.
209: comparting? Do you mean comparing?
Yes, we meant comparison, thank you. According to the note of Reviewer 1, we removed the discussion of bioinformatic pipelines.
293: State where in this study this information resides?
Thank you, this phrase appears too early in the text, before the real description of the experiment in Section 2.6. We have reformulated this sentence.
Not so minor:
310-330: This is prime text for removal as speculation without much point to the speculation.
We agree that the hypothesis is somewhat speculative and has no experimental proof yet. We deleted the detailed description.
